# Effect of Press Cake-Based Particles on Quality and Stability of Plant Oil Emulsions

**DOI:** 10.3390/foods13182969

**Published:** 2024-09-19

**Authors:** Tamara Schmid, Mathias Kinner, Luca Stäheli, Stefanie Steinegger, Lukas Hollenstein, David de la Gala, Nadina Müller

**Affiliations:** 1Zurich University of Applied Sciences, Institute of Food and Beverage Innovation, Einsiedlerstrasse 35, 8820 Wädenswil, Switzerland; tamaraschmid@bluewin.ch (T.S.); mathias.kinner@zhaw.ch (M.K.); luca.staeheli@bluemail.ch (L.S.); stefanie.steinegger@zhaw.ch (S.S.); 2Zurich University of Applied Sciences, Institute of Computational Life Sciences, Schloss 4, 8820 Wädenswil, Switzerland; lukas.hollenstein@zhaw.ch (L.H.); david_de_la_gala@yahoo.de (D.d.l.G.)

**Keywords:** Pickering emulsion, emulsion stability, regression model

## Abstract

Palm fat has uniquely optimal melting characteristics that are difficult to replace in products such as baked goods and chocolate-based items. This study investigates the efficacy of using Pickering emulsions derived from Swiss plant oils and their micromilled press cakes. Emulsification was carried out at both the lab and pilot scales using sunflower- and rapeseed-based recipes, with and without additional surfactants, for both oil-in-water and water-in-oil emulsions. The resulting emulsions were measured for viscosity and short- and long-term stability and linked to the properties of the raw materials. The results indicated that the contact angle, size, and macronutrient composition of the particles significantly impact emulsion quality, though differences in oil pressing methods might predominate these effects. The combination of particles and surfactants demonstrated a clear advantage with respect to interface stabilisation, with a suggested link between the wax content of the oil and particles and the resulting emulsion quality and stability.

## 1. Introduction

The quality of fats and oils significantly impacts the processability of food products such as baked goods and sweets. Palm fat has unique melting characteristics that are optimal for many products, providing both firmness at room temperature and rapid melting at slightly elevated temperatures, which is challenging to replace. This publication is part of a project in which the physical modification of plant oils through the production of particle-stabilised emulsions is tested as a means to mimic the properties of palm fat in cookie dough and chocolate products during processing, storage, and consumption. 

Pickering, or Ramsden, emulsions, which are emulsions where interfaces are stabilised by particles instead of surfactants, have been known since the early 20th century [1,2]. These emulsions have superior stability against coalescence and Ostwald ripening [3]. Particle stabilisation through particles with partial dual wettability is, in contrast to using classical emulsifiers, slower, but the adsorption of particles at the interface is considered extremely stable [4] if not irreversible [5,6], even for emulsions with larger droplet sizes [7]. The absence of traditional surfactants is advantageous, as the process is more environmentally friendly and cost-effective compared to using emulsifiers [8]. 

However, in order to achieve good interfacial stabilisation, the quality of the particle stabilisers themselves is crucial. Key qualities to consider include the concentration [5]; wettability, expressed as contact angle; size distribution; shape; and surface charge measured as zeta potential [9]. Particle sizes are required to be below 25 µm to avoid any gritty mouthfeel [10] and should be roughly ten times smaller than the droplet sizes [11,12]. 

The contact angle is crucial in determining whether an oil-in-water (O/W) or water-in-oil (W/O) emulsion is formed [13,14]. Mathematically, the contact angle can be described by Young’s equation; however, the surface of the particles must be “smooth, flat, homogeneous, inert, insoluble, non-reactive, non-porous, and of non-deformable quality” for the equation to be applicable [15]. Any deviation leads to difficulties in predicting the behaviour of particles in a Pickering emulsion [15]. Directly measuring powder contact angles is challenging due to their surface roughness and inhomogeneities [16,17], but it can provide a good indication of relative differences between particle properties, as shown by Rüegg et al. [18]. They demonstrated that while the contact angle measurements of particles pressed into dense tablets showed a weak correlation with the resulting foam bubble sizes, there was a medium-to-strong correlation between these measurements and the long-term stability of the foam, which was assessed by measuring how well the foam maintained its structure over time through drainage tests. According to Guo et al. [19], direct contact measurement is possible if particles are small enough and monodisperse. Maestro et al. [17] described alternatives to the direct measurement of contact angles, such as capillary rise experiments [20], the Washburn capillary rise method [21], or the gel trapping method [22]. While these contact angle measurement methods are well established for synthetic materials or well-defined natural materials such as silica particles, information on the applicability of these methods to natural raw materials is scarce [20].

Contact angles can be adjusted by the adsorption of surfactants or through chemical modification [23], but options for food-grade particles are limited. Gonzenbach et al. [6] discussed hydrophobisation through the adsorption of short-chain amphiphiles on particle surfaces for food industry applications. In addition, food-grade particles have been produced by coating inert silica particles with lecithin to increase hydrophobicity [24] (Eskandar et al., 2007). Lastly, hydrophobically modified cellulose fibres have also been shown to stabilise superstable Pickering foams [25]. Tavernier et al. [26] provided an overview of food-grade particles with the potential to serve as Pickering particles, including materials of biological origin like fats and wax crystals, protein–polysaccharide complexes, flavonoids, and protein particles [27]. An underutilised opportunity for more sustainable food production lies in using particles sourced from food side streams, which offer a wide range of particles with varying material qualities [3], such as press cakes from oil seeds. Sunflower or rapeseed press cakes, produced in large amounts worldwide, possess a good diversity of naturally occurring constituents [28], making them ideal candidates for use as particles in Pickering emulsions.

While particles lead to a particularly strong stabilisation of interfaces [4,5], their larger size compared to classic surfactants slows down interface stabilisation. Hence, combining particles with a fast-acting surfactant such as an emulsifier can be beneficial in continuous emulsion formation [29].

This study aims to elucidate the effect of varying qualities of sunflower and rapeseed oils and their respective press cake particles on emulsion quality and stability. Rapeseed and sunflower seeds are the raw materials of choice as the top two oilseeds grown in Switzerland. In addition, this study tests the impact of using combined surfactant–particle stabilisation systems in both oil-in-water and water-in-oil emulsions. It also compares production in batch mode with lower shear rates at the laboratory scale to continuous production with higher shear rates at the pilot scale. The corresponding tests were carried out in consecutive unifactorial designs. To conclude, the effects are summarised in correlation matrices to draw conclusions for follow-up work.

## 2. Material and Methods

### 2.1. Raw Materials

#### 2.1.1. Oils

Rapeseed oil, high oleic low linoleic (HOLL) quality, conventional cultivation, Grüninger AG, Mitlödi, Switzerland.Rapeseed oil, high oleic low linoleic (HOLL) quality, conventional cultivation, Florin AG, Muttenz, Switzerland.Sunflower oil, classic quality, biological cultivation, Grüninger AG, Mitlödi, Switzerland.Sunflower oil, high oleic (ho) quality, conventional cultivation, Florin AG, Muttenz, Switzerland.

#### 2.1.2. Press Cakes

Rapeseed press cake, high oleic low linoleic (HOLL) quality, conventional cultivation, Grüninger AG, Mitlödi, Switzerland.Rapeseed press cake, classic quality, conventional cultivation, Florin AG, Muttenz, Switzerland.Sunflower press cake, classic quality, biological cultivation, Grüninger AG, Mitlödi, Switzerland.Sunflower press cake, high oleic (ho) quality, conventional cultivation, Florin AG, Muttenz, Switzerland.The oils and press cakes were tested in the following combinations:Rapeseed oil HOLL, Grüninger AG with rapeseed press cake, HOLL, Grüninger AG, Mitlödi, Switzerland.Rapeseed oil HOLL, Florin AG with rapeseed press cake classic, Florin AG, Muttenz, Switzerland.Sunflower oil, classic from Grüninger AG with sunflower press cake classic from Grüninger AG, Mitlödi, Switzerland.Sunflower oil ho from Florin AG with sunflower press cake ho from Florin AG, Muttenz, Switzerland.

#### 2.1.3. Emulsifiers

Distilled monoglyceride (E471) (Dimonda HR, Danisco, Kreuzlingen, Switzerland).Polyglycerol polyricinoleate (E476) (PGPR, Esterchem, Staffordshire, UK).Polysorbate 20 (Tween20, Esterchem, Staffordshire, UK).Sunflower lecithin (SUNLEC M, Sime Darby, Zwijndrecht, The Netherlands).Rapeseed lecithin (LeciFis RAP TF IPM, Fismer Lecithin, Hamburg, Germany).

### 2.2. Production Methods

#### 2.2.1. Formation of Particle-Stabilised Emulsions

Particle-stabilised emulsions were created from four different oil–press cake combinations to compare the influence of various raw materials and their characteristics. The combinations used were as follows: rapeseed high-oleic low-linoleic (HOLL) press cake with its oil, rapeseed high-oleic low-linoleic (HOLL) press cake with classic rapeseed variety oil, sunflower high-oleic press cake with its oil, and organic sunflower classic seed variety press cake with its oil. These combinations were tested both at the lab scale (see Section 2.2.2) and the pilot scale (see Section 2.2.3). The process setup was based on a previous study on palm oil replacement in baked goods [30]. At the lab scale, each trial was repeated twice on different days, and each was analysed three times (n = 6). In comparison, pilot-scale tests were repeated three times on different days. The viscosity of the emulsions was measured three times (n = 9) directly after production, and their stability was quantified over time (1 h, 24 h, 168 h) using the emulsion index (see Section 2.3.2).

#### 2.2.2. Lab-Scale Emulsion Formation

To create emulsions at the lab scale, a formulation of 85% oil (*v*/*v*) and 15% water (*v*/*v*) was used for W/O emulsions, and 70% oil (*v*/*v*) and 30% water (*v*/*v*) was used for O/W emulsions. In both formulations, particles were added at a concentration of 5% by weight (*w*/*v*). Optionally, 0.5% by weight (*w*/*v*) of distilled monoglyceride (E471) (Dimodan HR, Danisco, Kreuzlingen, Switzerland) or polyglycerol polyricinoleate (E476) (PGPR, Esterchem, Staffordshire, UK) was added to the W/O emulsions. For sunflower-based O/W emulsions, 0.5% by weight (*w*/*v*) of polysorbate 20 (Tween20, Esterchem, Staffordshire, UK) or sunflower lecithin (SUNLEC M, Sime Darby, Zwijndrecht, The Netherlands) was optionally added. For rapeseed-based O/W emulsions, 0.5% by weight (*w*/*v*) of polysorbate 20 or rapeseed lecithin (LeciFis RAP TF IPM, Fismer Lecithin, Hamburg, Germany) was added.

The emulsions (in total 30 mL liquid phase) were generated by mixing oil and press cake in a 50 mL Falcon tube at a shear rate of 2199 s^−1^ for 60 s using a polytron (Polytron PT 2500 E, Kinematica AG, Buchs, Switzerland) with the dispersing geometry PT-DA 12/2EC-E157 (Kinematica AG, Buchs, Switzerland). Water was then added, and after waiting for an additional 60 s for phase separation, the mixture was homogenised at 9529 s^−1^ for 120 s using the same polytron. In cases where emulsifiers were added, they were dissolved in oil for W/O emulsions and in water for O/W emulsions and stirred at 2199 s^−1^ for 60 s prior to mixing oil and press cake. After emulsion formation, a drop test was conducted to ensure that the emulsion was the targeted O/W or W/O emulsion (see Section 2.3.2).

#### 2.2.3. Upscaling to Pilot Scale Using Rotor–Stator (RS) System and Rotating Membrane (RM) System

For pilot-scale production, the press cake particles (5% by weight (*w*/*v*) of the total water and oil content) were predispersed in 1/3 of the oil for 30 s at the highest speed using a mixer (GT800 Rotor, Lips AG, Uetendorf, Switzerland) for 30 s. The remaining 2/3 of the oil was then added using a polytron (MS1 CAA-R Ytron, F. Mundwiler & Co AG, Rüschlikon, Switzerland) and mixed for an additional 5 min at 7134 s^−1^. Next, this suspension of press cake particles in oil was transferred to the feed tank and stirred at a shear rate of 25 s^−1^ using an agitator (IKA GmbH RW 28 digital, Staufen, Germany). When using the emulsifier E471, 1/3 of the oil was first mixed with the emulsifier for 30 s using the mixer (GT800 Rotor, Lips AG, Uetendorf, Switzerland) at the highest speed before predispersing with the press cake particles. Subsequently, the remaining steps were the same as without emulsifiers.

With particles already suspended in the oil phase, the rotor–stator pilot plant system (Megatron MT-FM 50 Pilotplant, Kinematica AG, Lucerne, Switzerland) was adjusted to the following ratio: 85.6% oil–press cake suspension and 14.4% water at a total throughput of 26.3 L/h to produce a water-in-oil emulsion. A shear rate of either 47,574 s^−1^, 71,314 s^−1^, or 95,086 s^−1^ was set, and the relative pressure in the emulsification head was maintained at 1 bar.

### 2.3. Analysis Methods

#### 2.3.1. Characterisation of Press Cakes

To evaluate the influence of the particles on emulsion quality, different analyses were conducted. Part of the analyses were performed in external accredited laboratories: (i) mean particle size [µm] and particle distribution width (×90.0/×10.0) by means of microscopic analysis according to ISO standard 13320:2020 [31], (ii) BET surface particle density [g/cm^3^] using helium pycnometry, and (iii) nutrient analysis, including protein content [g/100 g], fat content [g/100 g], water content [g/100 g], and total dietary fibre content [g/100 g].

In addition, the contact angle [°] was analysed at the Zurich University of Applied Sciences by pressing particles into dense tablets using a tablet press (MTQX-1, GlobePharma, Monmouth Junction, NJ, USA), adding a defined droplet of water to the surface, and quantifying the contact angle (Contact Angle System OCA Goniometer, Dataphysics, Filderstadt, Germany) (n = 16). Lastly, light microscope images were taken using an inverted light microscope (RLV-100-G, Discover Echo Inc., San Diego, CA, USA). The selection of analyses of particles was based on a previous study on particle-stabilised foams [18].

#### 2.3.2. Analysis of Emulsions

Drop test

A drop test was conducted to determine whether the emulsion was oil-in-water (O/W) or water-in-oil (W/O). One drop of the emulsion was added to a bowl filled with water, and another was added to a bowl filled with oil using 5 mL disposable pipettes. If the emulsion drop dissolved the water, it was classified as an O/W emulsion. If the emulsion drop dissolved in the oil, it was classified as a W/O emulsion. The drop test was performed after each lab-scale emulsion formation to verify the type of emulsion produced.

Emulsion stability or Emulsion Index

The emulsion stability was determined by filling 14.5 mL of the prepared emulsions into 15 mL Falcon tubes and storing them at 20 °C. The volumes [ml] of the cream layer (oil phase), the emulsion layer (emulsion), and the serum layer (water phase) were determined relative to the initial volume [mL] via visually assessing the ml scale of the tubes at 1 h, 24 h, and 168 h after the emulsion formation, respectively, according to the protocol described by McClements [32]. The emulsion index EI [%] is defined by
(1)$$EI_{t}=100\cdot \frac{V_{t}}{V_{0}}$$
where V_0_ is the total volume directly after emulsion formation, and V_t_ is the volume without any phase separation after storages of *t* = 1 h, 24 h, and 168 h, respectively.

Viscosity measurement

The viscosity of the emulsions as a function of the shear rate was measured immediately after production at both the lab scale (n = 6) and pilot scale (n = 9) using a rheometer (MCR 702, Anton Paar AG, Graz, Austria). To perform the measurements, 19 mL of the emulsion was transferred to a sample cup and analysed with a coaxial cylinder measuring system (CC27-SN71788, 64353, Anton Paar AG, Graz, Austria). A logarithmic shear ramp was run at 20 °C from 1 to 10 s^−1^ measuring 10 points each for one second, from 20 to 100 s^−1^ measuring 9 points each for one second, from 200 to 1000 s^−1^ measuring 9 points each for one second, and from 2000 to 3000 s^−1^ measuring 2 points each for one second.

### 2.4. Experimental Design

First, the effect of different raw material combinations (oilseed variety, type of oil, type of press cake) on the resulting emulsion index was compared at the lab scale in batch mode production. Analysis was carried out after 1 h, 24 h, and 168 h of storage at 20 °C.

Second, based on the results of the first step, the effect of an additional emulsifier on the emulsion index was tested for selected recipes and compared after 1 h, 24 h, and 168 h of storage at 20 °C following lab-scale emulsion production.

Subsequently, the experimental setup was scaled to pilot-scale emulsification equipment working in continuous mode. Here, the effects of different recipe combinations (with and without press cake as well as with and without emulsifier) were compared. The emulsion index was observed at 1 h, 24 h, and 168 h after production and storage at 20 °C, while the viscosity was measured directly after production.

The results obtained from each of these unifactorial designs were analysed by means of a Kruskal–Wallis test (α = 0.05) for equal means of the groups according to the independent variables, followed by a pairwise post hoc test, specifically an unpaired Wilcoxon test (α = 0.05). Significant differences were indicated with different letters (compact letter display) where only results with no overlap in letters differ significantly from each other (e.g., a and b are significantly different, whereas a and ab are not significantly different). For the calculations and visualisations, the following software packages were used: R (version 4.2.1) and R Studio (version 2023.09.0) with the Hmisc (version 5.1-2), spatstat (version 3.0-8), multcompView (version 0.1-10), and ggplot2 (version 3.5.0) packages.

Lastly, the relationships between the properties of the emulsions and the characteristics of the stabilising particles were quantified through their Pearson correlations and visualised using kernel density estimation for each pair of variables. To construct a coherent dataset for this analysis, the viscosities and emulsion indices (at each time point analysed) were averaged over the three measurements per repetition, as only the repetitions were identifiable over time.

The Pearson correlation coefficients and p-values between the means were computed pairwise using the SciPy package (version 1.13.0) in Python (version 3.12.1). Pairwise scatterplots and probability densities were produced with the Seaborn package (version 0.13.2), which internally uses SciPy to compute a kernel density estimate using Gaussian kernels with the default bandwidth following Scott’s rule [33].

## 3. Results and Discussion

### 3.1. A Particle Analysis of the Raw Material

The compositional analysis (Table 1) clearly shows that the two rapeseed press cakes have higher protein and lower dietary fibre contents compared to the sunflower press cakes. This aligns with findings from Lomascolo et al. [34], which reported similar values and trends. The fat content is particularly high in the sunflower classic press cake. This high fat content could be attributed to the organic quality of the press cake, which allows for a maximum pressing temperature of 50 °C and is significantly lower than standard oil pressing [35]. A review by Savoire et al. [36] highlights that testing different varieties of specific oilseeds shows that the variety has a higher impact on the resulting oil pressing performance and yield than can be explained merely by the raw material composition, such as the oil content. One of the explanations that was found for this phenomenon is provided by Zheng et al. [37], who show that for flaxseed, the missing link between variety and resulting oil yield can be found in the microstructure, specifically in the thickness of the hulls.

The particles from the different press cakes are comparable in density and size (Table 2). The mean particle sizes (×50.0) are relatively large for the effective particle stabilisation of interfaces, as particles should typically be tenfold smaller than the droplets they stabilise [11,12]. However, achieving such small particles from fibrous materials such as press cakes is challenging. Fibrous particles can form network aggregates that entrap water, thereby contributing to the stability of oil-in-water emulsions by increasing the viscosity of the aqueous phase [38]. Shapewise, particles from both rapeseed press cakes were more roundish, while the sunflower particles consisted of a mixture of roundish particles interdispersed by a few more fibrous and elongated particles (Figure 1).

The two sunflower press cakes exhibit significantly larger contact angles compared to the rapeseed press cakes, as confirmed by a Kruskal–Wallis test (*p* < 10^−8^) and a Wilcoxon pairwise post hoc test (*p* < 0.0002 between all sunflower and rapeseed groups and *p* > 0.1 within each of the sunflower and rapeseed groups, respectively). For both high-oleic and classic varieties, the contact angle is close to 90°, which is considered optimal for stabilising two-phase systems [39]. A contact angle close to 90° indicates that more energy is needed to detach the particles from the interface. Typically, contact angles ≤ 90° are optimal for oil-in-water emulsions [40,41], while hydrophobic particles with contact angles ≥ 90° are ideal for water-in-oil emulsions.

### 3.2. Influence of Raw Materials on W/O Emulsion Stability at Lab Scale

#### 3.2.1. Influence of Oil and Press Cake Quality on Stability of Particle-Stabilised W/O Emulsion at Lab Scale

The emulsion index of W/O emulsions produced at the lab scale (Figure 2) from sunflower oil and press cake from the classic variety is higher than for all other emulsions at 1 h after production. However, the loss in the stability of these emulsions over time decreased more rapidly than for all other recipes. Consequently, this positive effect was not observable after 24 h of storage, and the emulsion index of this recipe became lower than that of all other recipes after 168 h. At 1 h, there was no significant difference in the emulsion index between the two rapeseed-based recipes and the sunflower recipe with high oleic oil and press cake, although there was a tendency for the rapeseed-based recipes to have a lower emulsion index. After 168 h of storage, the opposite effect was observed: while the emulsion indices of the rapeseed-based recipes and the sunflower recipe with high oleic oil and press cake showed no significant difference, the indices of the rapeseed-based recipes were slightly higher.

The highest initial emulsion index for the emulsion produced with particles from the sunflower classic variety aligns with theoretical expectations. This is because only these particles had contact angles greater than 90°, which is linked to a better stabilisation of water-in-oil emulsions. With regard to the shape of particles, both roundish and elongated shapes were found to stabilise emulsions [13].

Several factors might have contributed to the observed poor long-term stability of these emulsions. It is possible that the high energy required for absorption at the interface, which is necessary at contact angles close to 90°, was not achieved during emulsification. Consequently, sufficient particle stabilisation was not achieved [17].

Despite the small mean particle size (×50.0) of about 5 µm for all particles (see Table 2), agglomeration effects were observed. If these particle clusters were not adequately dispersed during emulsification, only part of the interface might have been covered by particles, hence making it impossible to stabilise the interface properly. Zhao et al. [42] report that even a configuration where particles are aggregated in some domains while the rest of the emulsion droplet surface is not covered by particles has the potential to stabilise the interface. In addition, the droplet size in emulsions (Appendix A) ranged from about 1 to 10 µm, with a few larger droplets up to about 100 µm. This made the particles relatively large in comparison, which hindered effective interphase stabilisation for the smaller droplets [11,12]. Gould et al. [43] tested Pickering-stabilised oil-in-water emulsions using cocoa particles and showed that natural food particles are well suited to stabilise this type of emulsion. 

#### 3.2.2. Influence of Emulsifiers on Stability of Particle-Stabilised O/W and W/O Emulsions at Lab Scale

For water-in-oil emulsions (Figure 3), the addition of emulsifiers led to higher emulsion stability at 1 h for the sunflower recipe, with the highest stability observed in the recipe with PGPR. No significant change in stability was found in the rapeseed-based recipes. Over a 168 h observation period, the sunflower recipe with E471 was more stable than the emulsions with PGPR and without emulsifier. For the rapeseed-based water-in-oil emulsions, no significant difference in stability was found after 24 h, with the emulsion without emulsifier being slightly more stable than those with emulsifier, albeit at very low emulsion indices overall.

The results for the oil-in-water emulsions showed 100 percent stability after 1 h of storage for all recipes. After 24 h, all the recipes with emulsifiers were more stable than those without, with the emulsions containing Tween20 remaining completely stable, maintaining an emulsion index of 100%. After 168 h, similar trends were observed, with the exception that the rapeseed-based recipe with lecithin did not show increased stability compared to the recipe without emulsifier. Both the rapeseed- and the sunflower-based recipes with Tween20 demonstrated the highest emulsion stability, with a mean emulsion index of 95% at 168 h of storage.

The good stability at 1 h and acceptable stability after 24 h for all the oil-in-water emulsions without emulsifiers suggest that the particles are better suited for oil-in-water than for water-in-oil emulsions. This aligns with the expectations based on the contact angles shown in Table 2 for the rapeseed particles. For the sunflower particles, a contact angle slightly above 90° was measured, indicating a tendency to favour the formation of water-in-oil emulsions. However, the mean contact angle was very close to 90° (90.58°) with a standard deviation of 4.22. While some authors found that the direct measurement of contact angles in powders is possible [18,19], the majority suggested alternative measurement methods for powdered solids [16,17] and considered direct measurement unreliable. Furthermore, the particle size distribution can influence the particle stabilisation efficiency, as tests by Gould et al. [43] on oil-in-water emulsions stabilised through cocoa particles showed. Gould et al. found that small particles stabilised the droplet interface, while larger particles in the size range of the droplets contributed to the structure of the continuous aqueous phase. They furthermore reported an advantageous effect of the removal of water-soluble molecules before use. Lastly, several studies on the effect of the shape of particles on their ability to stabilise O/W and W/O emulsions were compared by Gonzalez Ortiz et al. [13] and showed that some roundish particles such as silica or Polystyrene can stabilise both O/W and W/O emulsions, while others such as TiO_2_ and Fe_2_O_3_ were only applicable in O/W emulsions. For fibrous-like particles, only O/W stabilisation was reported. While none of the literature on rapeseed and sunflower particles and their ability to stabilise interfaces is available to date, the fraction of elongated particles found in micromilled sunflower press cake might have led to the higher stability of sunflower-based O/W emulsions over the short and long term compared to W/O emulsions. 

The successful use of glycerol monostearate as an emulsifier for water-in-oil emulsions was demonstrated by both El-Aooiti et al. [44] and García-Ortega et al. [45]. El-Aooiti applied the emulsifier in the form of nanosized colloidal particles and found a high stability against demulsification [44]. García-Ortega et al. [45] applied the water-in-oil emulsification step as the first step in producing oleogels with ethyl cellulose, performing the emulsification step at elevated temperatures of 70 °C to avoid early crystal formation. Our results with glycerol monostearate (E471) were less satisfactory in terms of emulsion stability, suggesting that a prolonged heating step, as applied by García-Ortega et al. [45], is of critical importance.

PGPR, as a ‘classic’ surfactant used in water-in-oil emulsions, is widely discussed in the literature, especially in the context of soybean oil-based emulsions [46]. Pan et al. [47] tested mixtures of soyasaponin and PGPR in water-in-corn oil emulsions and found optimal water-to-oil ratios of 30 to 70%, which is higher than the percentages used in the current study (15% water to 85% oil). The addition of 3% PGPR and 1% soyasaponin increased emulsion stability, showing a synergistic effect between the two additives. A direct comparison with our results is difficult due to the higher dosages used in Pan et al.’s trials. Ribas-Fonseca et al. [48] found highly promising long-term stability results for ‘high-internal-phase’ sunflower oil-based water-in-oil emulsions with water contents up to 80%, stabilised with a combination of sunflower wax and PGPR. However, stabilisation purely with PGPR led to unsatisfactory results. While this approach was out of the scope of the current work, it emphasises the importance of phase fractions on emulsion stability and the potential of approaches combining surfactants with particles.

Limitations in the use of pure lecithin as an emulsifier for water-in-oil emulsions, such as flocculation [46] or phase inversion [49], are well documented, with efforts being made to improve its effectiveness through either the addition of particles such as zein [49] or polysaccharides as thickening agents [50]. These limitations were clearly visible in our trials as well. In contrast to the limitations of lecithin, findings by Prodromidis et al. [51] demonstrate the high potential of Tween20 in emulsifying water-in-oil emulsions. In combination with monoglycerides, structured emulsions were achieved at low temperatures, with crystals forming at the droplet interface to act as Pickering particles [51].

Overall, the oil-in-water emulsions, both with and without emulsifiers, proved to be much more stable than the water-in-oil emulsions. Nevertheless, they were not considered in further trials due to the elevated water activity of oil-in-water emulsions, which is undesirable for the target applications of this project.

Based on its slightly better long-term stabilisation performance, glycerol monostearate (E471) was chosen for water-in-oil emulsions for further trials.

### 3.3. Upscaling Using a Rotor–Stator System

#### 3.3.1. Influence of Different Shear Rates on Emulsion Quality

Water-in-oil emulsions produced with the rotor–stator device at shear rates of 47,574 s^−1^, 71,314 s^−1^, and 95,086 s^−1^ were analysed for their stability. After 1 h of storage, the sunflower-based recipe with high-oleic sunflower oil showed a mean emulsion stability of 77.78 ± 13.14% at a shear rate of 47,574 s^−1^, 78.16 ± 11.95% at 71,314 s^−1^, and 88.89 ± 4.81% at 95,086 s^−1^. The rapeseed-based recipe with HOLL rapeseed oil had comparable emulsion stability but higher standard deviation at a shear rate of 47,574 s^−1^, with 72.41 ± 26.26%. At 71,314 s^−1^, the emulsion stability of the rapeseed-based emulsion was exceptional, with 96.55 ± 2.99% and very small standard deviation, while at 95,086 s^−1^, the emulsion index was slightly lower at 95.98 ± 1.22%, but the reproducibility of the results was high. A clear trend towards a higher emulsion index with an increasing shear rate was observed for both sunflower- and rapeseed-based emulsions, with the maximum stability being reached at lower shear rates for rapeseed-based emulsions than for sunflower-based emulsions. Generally, the standard deviations were smaller for all emulsion indices produced at higher shear rates. The observed recipe-dependent optimal parameter settings are often noted phenomenologically in emulsification trials due to competing positive mechanisms, such as higher disrupting forces acting during emulsification, and negative effects, such as rising temperatures at higher shear rates and increased droplet collisions in the turbulent flow field in the emulsification area of the rotor–stator head [52]. Based on these findings, all further continuous trials were conducted at the highest shear rate of 95,086 s^−1^.

#### 3.3.2. Influence of Raw Materials on Emulsion Quality at Pilot Scale

As higher shear rates produced emulsions with higher stability and lower standard deviation (Figure 4), the subsequent raw material tests were only conducted at the highest shear rate of 95,086 s^−1^. Compared to lab-scale trials, this led to a tenfold increase in effective shear rates (from 9529 s^−1^ with the polytron to 95,086 s^−1^ with the rotor–stator device), resulting in a significant increase in short-term emulsion stability (1 h storage time) for all recipes. The rapeseed-based recipe with classic oil and HOLL press cake plus emulsifier (E471), and both sunflower-based recipes with emulsifier, showed higher stability than the corresponding emulsions without emulsifier. For the HOLL rapeseed-based recipe, a slight but not significant increase in emulsion stability after 1 h was observed for the emulsion with E471.

The positive effect of combined emulsifier–particle-stabilised emulsion systems has been analysed in numerous studies and is linked to their different interfacial properties, adsorption kinetics, and chemical reactivities (e.g., [53,54]). Nesterenko et al. [55] found a strong effect of surfactant concentration in dual emulsifier–particle-stabilised water-in-oil emulsions when testing dosages between 0.1% and 1.8%, with the highest stabilities at low surfactant concentrations of 0.1%. While the current trials used relatively low dosages of 0.5% emulsifier, a further reduction might be advisable in future studies. However, while the low dosage was positive for emulsion stability, the higher dosage was strongly favourable for minimising droplet size [55].

The emulsions produced at the pilot scale (Figure 4) without emulsifier showed comparably low medium- and long-term stability over 24 and 168 h to those produced at lab scale (Figure 2), considering the tenfold higher shear rates. 

After 24 and 168 h of storage, all emulsions with emulsifier had significantly higher stabilities than the corresponding emulsifier-free emulsions. No clear trends between different types of rapeseed and sunflower varieties could be deduced from the data in Figure 3. The long-term stability of emulsions with emulsifier was higher at the pilot scale (Figure 4) than at the lab scale (Figure 3).

Microscope images (Appendix A) were not clear enough to allow for an interpretation of exact differences in droplet sizes. However, an interesting observation is the disappearance of large press cake fragments in samples produced at higher shear rates (pilot-scale trials), which resulted in an opaque overall image quality. In addition, droplet shapes were not uniformly circular in emulsions produced at the lab scale but instead had multiple shapes. These findings suggest that the long-term stability of emulsions produced at higher shear rates at the pilot scale might be supported by an increase in the viscosity of the continuous phase through the dissolution of larger particles. This theory would need to be tested by comparing the viscosity of emulsions produced at the lab and pilot scales in follow-up trials.

#### 3.3.3. Impact of Recipe on Emulsion Viscosity at Pilot Scale

Figure 4 shows the effect of different raw material combinations on the resulting water-in-oil emulsion viscosity at different shear rates, compared to pure non-emulsified oils. The results clearly show that all emulsions are significantly higher in viscosity than the pure oils, regardless of whether the emulsification was only conducted with press cakes or with press cakes and emulsifiers and regardless of the shear rate applied to the emulsions. Interestingly, organic oil had the lowest viscosity in oil form but the highest viscosity after emulsification with sunflower press cake particles from organic quality at low shear rates, an effect that was no longer observed at high shear rates. The lower oil viscosity of pure organic sunflower oil can be attributed to the particularly high content of the unsaturated fatty acid ‘linoleic acid’ compared to the other oils (63.6% compared to a range of 9.3% to 9.8% for all other oils). The higher viscosity at low shear rates might be explained by a higher wax content in the oil, either through natural differences between oilseed varieties or achieved through processing (e.g., lower pressing temperatures or the absence of winterisation in organic oil) [35]. Of the four oils, three were translucent at room temperature, with only the classic sunflower oil in organic quality being opaque (Appendix A), suggesting a much higher wax content in the latter.

While the viscosity of pure oils remains nearly constant at different shear rates, the viscosities of all emulsions decrease with increasing shear rates (Figure 5). This behaviour is typical for emulsions, where droplets deform under shear, and an alignment of both the deformed droplets and the press cake particles leads to a reduction in the emulsion’s viscosity [55]. Dietary fibres support both network formation (insoluble dietary fibres) and swelling (soluble fraction), which can positively affect viscosity at low shear. With increasing shear, an alignment of the fibres typically occurs, resulting in a shear-thinning effect, as shown for other fibre-rich side stream materials by Bonarius et al. [56].

Lastly, it was observed that at a low shear rate of 1 s^−1^, the viscosity of all emulsions without emulsifier was greater than that of the emulsion with emulsifier, while the opposite effect was observed at higher shear rates. This might indicate that in the combined emulsifier–particle stabilisation, the interface is primarily stabilised by the emulsifier, and the particles largely contribute to increasing the viscosity of the surrounding matrix. However, no positive synergistic effect of the combination of surfactant and particles, as obtained by Nesterenko et al. [55] at low surfactant dosages, was observed. No explanation has been found so far as to why the viscosity of the combined emulsifier–particle-stabilised system is lower at high shear rates than the system containing particles only. While the rheological behaviour and the effect of surfactant-to-particle ratios in concentrated emulsions with oversaturated dispersed phases are well researched (e.g., [57]), the observed effects cannot be transferred to more diluted emulsions with lower amounts of inner phase fractions, as produced here. One potential explanation might be the adsorption of the surfactant to the particle surface, as observed by Nesterenko et al. [55], which might influence the behaviour of the entire emulsion under shear.

The measurement of palm oil samples was not possible using the same protocol, as the viscosity was too high and the rheometer stopped; hence, no comparison is provided here.

#### 3.3.4. A Correlation Matrix Regarding the Influence of the Raw Material Characteristics on the Resulting Emulsion Quality

The influence of particle characteristics and nutritional content on the viscosity and stability of the emulsions was investigated through their linear correlations. To reduce variance and increase interpretability, this analysis focused on the W/O emulsions produced with the rotor–stator system. Pairwise scatterplots and kernel density estimates of the joint probabilities, as well as the Pearson correlation coefficients and respective *p*-values, were computed: particle characteristics versus emulsion properties are shown in Figure 6, and particle nutritional composition versus emulsion properties are shown in Figure 7.

From Figure 6, it is evident that the contact angle is moderately negatively correlated with the emulsion viscosity and moderately positively with the longer-term emulsion stability (emulsion index after 168 h). These correlations are essentially driven by the sunflower-based recipe with classic press cake, which leads to lower viscosities but higher stability at larger contact angles. Further, the particle size distribution is moderately to marginally correlated with the emulsion index at intermediate storage times (24 h), with smaller particles (x_50.0_) leading to slightly more stable emulsions and wider particle size distributions (x_90.0_/x_10.0_) leading to slightly more stable emulsions. However, this result must be taken only as a hint, as the differences between the particle size distributions could not be statistically analysed in this study (see Table 2 and discussion thereafter). Visual observation (Appendix A) suggests a higher wax content in the classic sunflower oil that was used in combination with its press cake particles. These waxes in the oil phase might have further contributed to the different behaviour observed for emulsions produced from classic sunflower oil.

The higher stability of emulsions with smaller stabilising particles is consistent with findings from Costa et al. [11] and Dickinson [12], which showed that particles approximately tenfold smaller than the droplets are optimal for stabilising droplets. As the particles in these trials are relatively large, any reduction in size was expected to improve emulsion stability. The positive correlation between contact angle and emulsion stability aligns with expectations, as the contact angles ranged from 78.91° ± 3.51 to 90.58° ± 4.09, with the classic sunflower press cake having the highest contact angle (>90°), which is optimal for stabilising water-in-oil emulsions [4,40].

Concerning the particle nutritional composition, Figure 6 indicates that the emulsion viscosity is strongly correlated with protein, fat, and fibre content, increasing with protein and decreasing with the fat and fibre content of the particles. Again, the classic sunflower press cake-based recipe appears to drive these correlations. While the emulsion index at intermediate storage times (1 h, 24 h) is uncorrelated with the macronutrient composition of the particles, the emulsion’s longer-term stability is moderately to strongly correlated with protein (negative), fibre (positive), and fat content (positive). Regarding the protein and dietary fibre content, an increasing dietary fibre content would typically be expected to increase the emulsion viscosity, and an increasing protein content would typically enhance emulsion stability. However, the findings in Figure 6 show the opposite effect. The observation that emulsion viscosity decreased with the increasing dietary fibre content of the particles might indicate that part of the water phase was absorbed by the dietary fibres, resulting in a volume decrease in the inner phase of the emulsion, which in turn lowers the emulsion viscosity [58]. An alternative explanation could be that the observed correlations are artefacts of an as-yet-unknown factor. As the correlation is driven by the classic press cake from organic quality, which was pressed at much lower temperatures and hence contains more residual oil and presumably a higher wax content [35], it would be advisable to assess the press cake composition in more detail to understand differences between this and other press cakes, such as wax content. Unfortunately, established wax analysis methods do not allow for measurement in solids. Moreover, the classic sunflower oil appeared to contain more waxes visually (Appendix A). 

## 4. Conclusions

While the use of particles to stabilise emulsions is promising, the results showed that pure particle stabilisation with oil press cake particles was insufficient for water-in-oil emulsions, necessitating the addition of emulsifiers. Conversely, oil-in-water emulsions stabilised solely with particles exhibited good medium-term stability.

Comparing different recipes and production methods highlighted that while particle contact angle and size impacted emulsion quality, the effect was limited due to relatively small differences in attributes between different press cakes. Using natural raw materials thus had the disadvantage of limiting the variability in the raw material properties. For future trials, it is advisable to consider additional particles that differ more significantly from each other. Regarding raw material composition, dietary fibre, protein, and fat content were found to influence emulsion properties. However, this correlation was mainly driven by one of the press cakes where a gentler pressing protocol was used, resulting in lower temperatures during pressing, higher residual oil, and potentially higher wax content. This might contribute to interphase stabilisation [29]. In addition, the press cakes were combined with their respective oils, and based on a visual observation of the oils’ transparency, the wax content of the classic sunflower oil appeared higher than all other oils. Therefore, for future trials, it is advisable to use raw materials processed similarly, such as comparing different bio-quality raw materials, as pressing protocols in organic quality differ from standard oil pressing.

Based on the findings and the goal to produce sustainable natural alternatives to palm fat from local plant oils, maximising interphase stabilisation using oil with high wax content is suggested for water-in-oil emulsions. For oil-in-water emulsions, using finely milled press cake from the same oil plants appears to be a promising approach.

## Figures and Tables

**Figure 1 foods-13-02969-f001:**
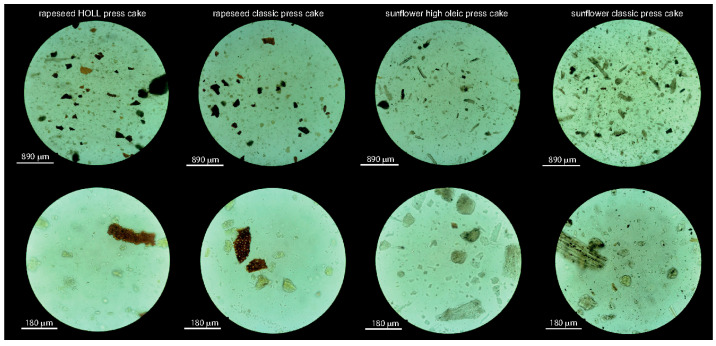
Light microscopy images of particle from rapeseed and sunflower press cakes dispersed in oil.

**Figure 2 foods-13-02969-f002:**
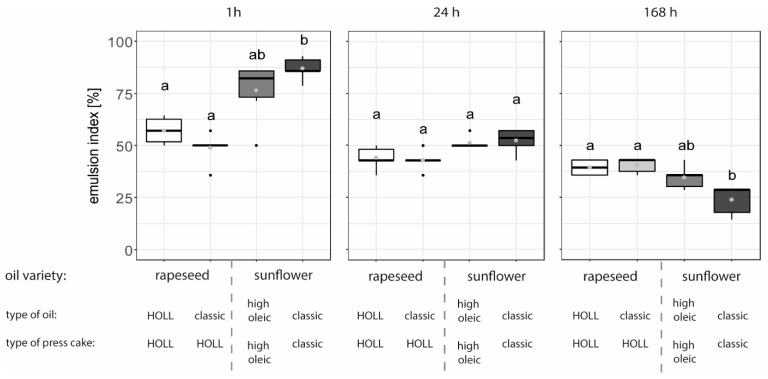
The emulsion index [%] as a measure of the emulsion stability of water-in-oil emulsions consisting of different raw material combinations produced at the lab scale and observed over 1 h, 24 h, and 168 h storage at 20 °C (n = 6). Letters (a, ab, b) indicate groups of significance, i.e., group means not sharing any letter are significantly different by the Wilcoxon test with a *p*-value of 0.05.

**Figure 3 foods-13-02969-f003:**
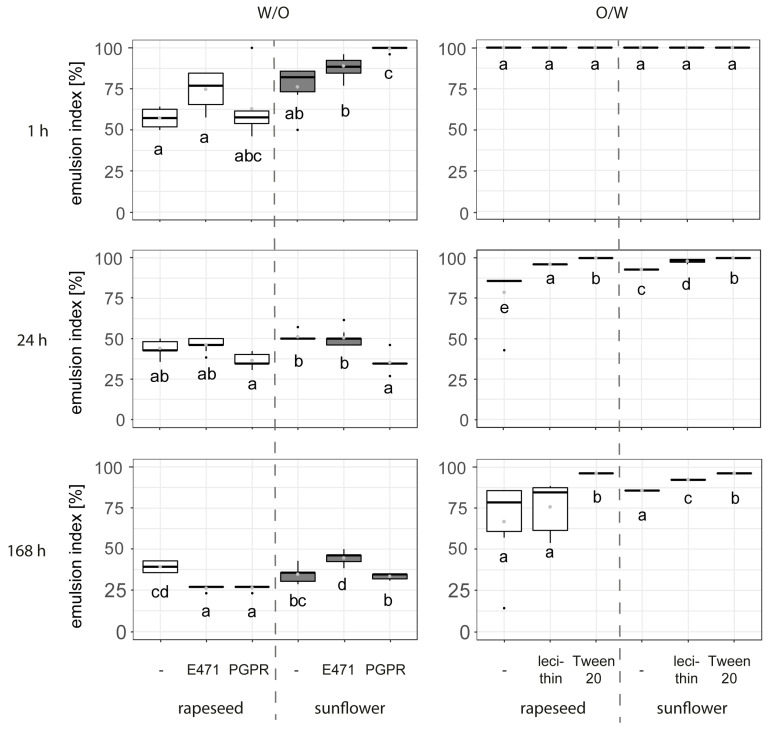
The emulsion index [%] of water-in-oil versus oil-in-water emulsions with different raw material combinations with and without emulsifiers (0.5% of either E471 or PGPR for water-in-oil emulsions, 0.5% of either lecithin or Tween20 in oil-in-water emulsions) produced in a lab-scale device observed at 1 h, 24 h, and 168 h of storage at 20 °C. All rapeseed-based recipes consisted of HOLL oil and HOLL press cake, and all sunflower-based recipes consisted of oil and press cake from a classic sunflower variety (n = 6). Letters (a to e) indicate groups of significance, i.e., group means not sharing any letter are significantly different by the Wilcoxon test with a *p*-value of 0.05.

**Figure 4 foods-13-02969-f004:**
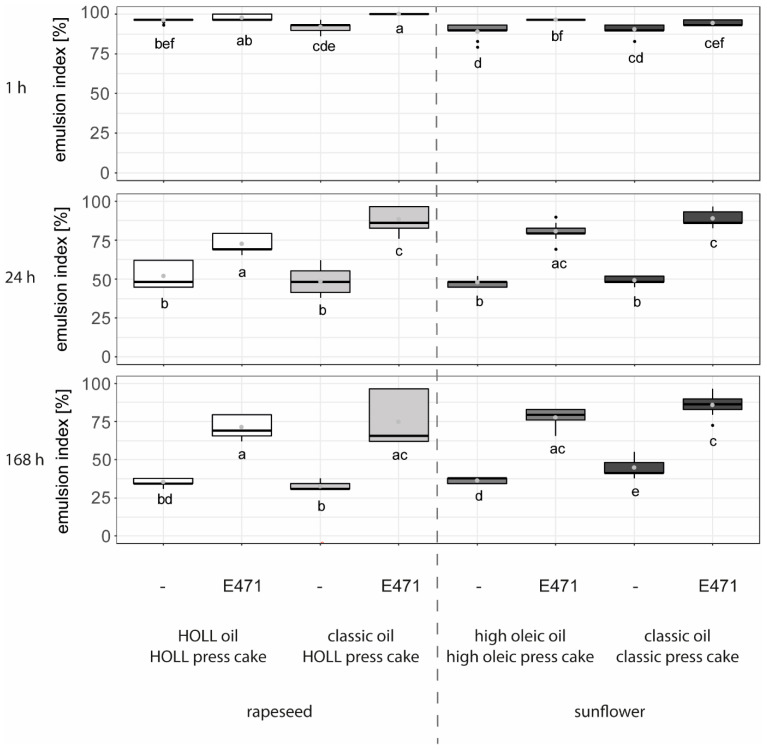
The emulsion index [%] of water-in-oil emulsions from different raw material combinations, with and without emulsifier (E471), produced on a pilot-scale rotor–stator device (Kinematica Malters, Switzerland) and observed over time (1 h, 24 h, and 168 h storage at 20 °C, n = 9). Letters (a to f) indicate groups of significance, i.e., group means not sharing any letter are significantly different by the Wilcoxon test with a *p*-value of 0.05.

**Figure 5 foods-13-02969-f005:**
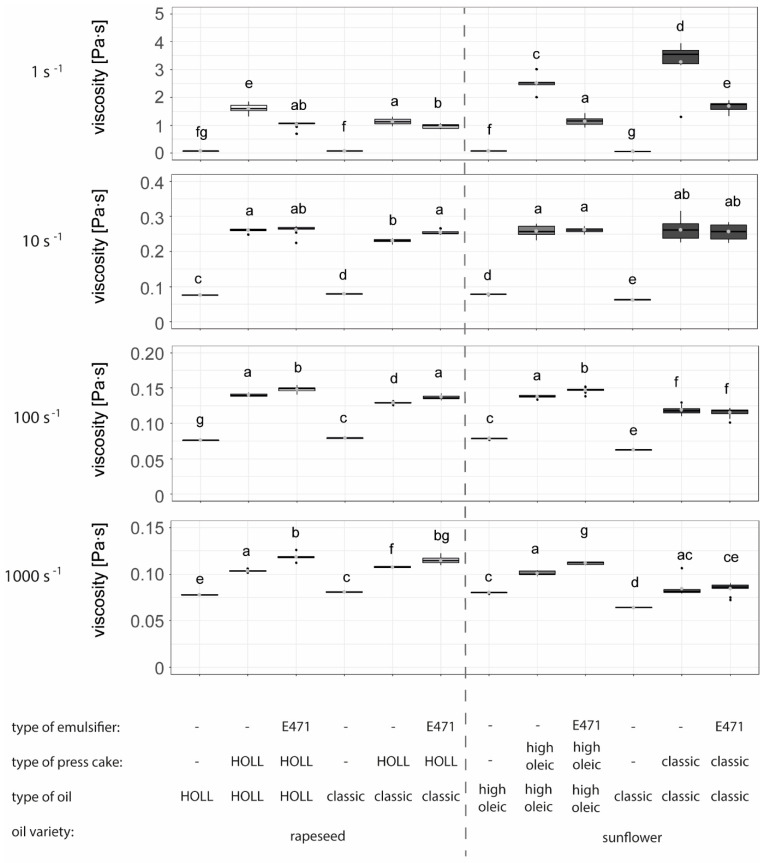
The viscosity [Pa·s] of water-in-oil particle-stabilised emulsions with and without emulsifiers compared to pure oils measured at different shear rates (1 s^−1^, 10 s^−1^, 100 s^−1^, 1000 s^−1^) produced on a rotor–stator device (Kinematica, Malters, Switzerland) directly after production (n = 9). Letters (a to g) indicate groups of significance, i.e., group means not sharing any letter are significantly different by the Wilcoxon test with a *p*-value of 0.05.

**Figure 6 foods-13-02969-f006:**
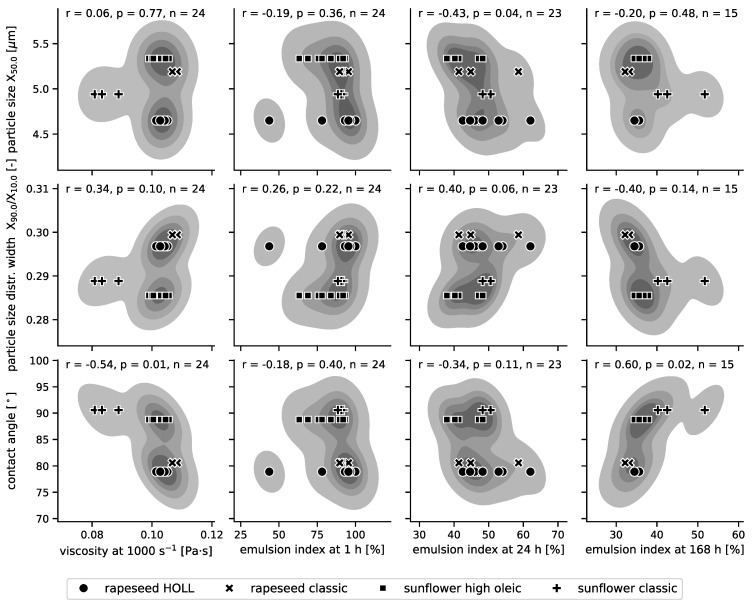
A correlation matrix between the particle properties (mean size [µm], size distribution width [-], and contact angle [°]) and the emulsion viscosity measured at a shear rate of 1000 s^−1^ [Pa·s], as well as the emulsion index [%] at 1 h, 24 h, and 168 h after production. Pairwise scatterplots are shown on top of the contours (5 levels) of a kernel density estimate of the joint probability density. Pearson correlation coefficients r, respective *p*-values, and the number of observations n are shown for each pair. Only W/O emulsions produced at the pilot scale and stabilised with 5% particles using the rotor–stator system without the addition of emulsifiers were analysed.

**Figure 7 foods-13-02969-f007:**
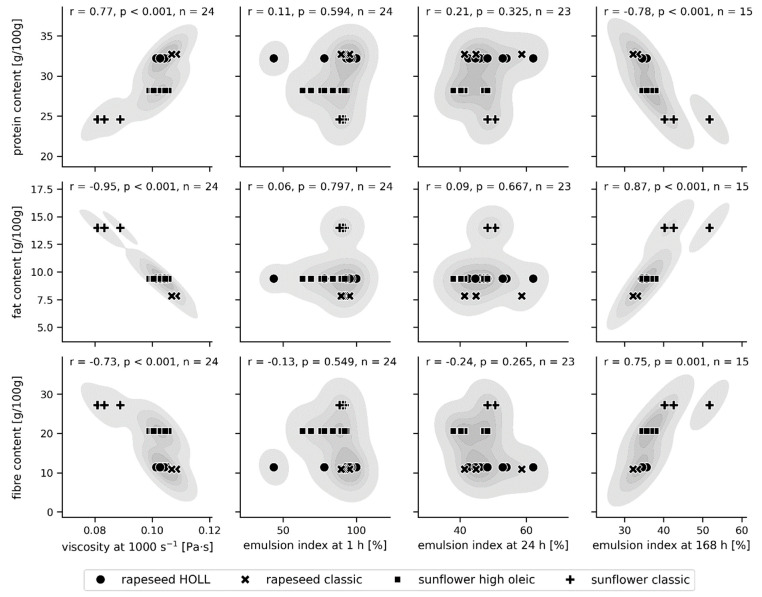
A correlation matrix between the particle nutrition content (protein [g/100 g], fat [g/100 g], and fibre [g/100 g]) and the emulsion viscosity at a shear rate of 1000 s^−1^ [Pa·s], as well as the emulsion index [%] at 1 h, 24 h, and 168 h after production. Pairwise scatterplots are shown on top of the contours (5 levels) of a kernel density estimate of the joint probability density. Pearson correlation coefficients r, respective *p*-values, and the number of observations n are shown for each pair. Only W/O emulsions produced at the pilot scale using the rotor–stator system without the addition of emulsifiers were analysed.

**Table 1 foods-13-02969-t001:** The composition of particles including fat, protein, fibre, mineral, and water content analysed in external accredited laboratories for the following samples: rapeseed press cake from HOLL rapeseed and conventional cultivation, rapeseed press cake from classic varieties and conventional cultivation, sunflower press cake from high-linoleic variety and conventional cultivation, and sunflower press cake from classic variety and organic cultivation. The results are given in mean values plus standard deviations.

	Fat Content [g/100 g]	Protein Content [g/100 g]	Fibre Content [g/100 g]	Mineral Content [g/100 g]	Water Content [g/100 g]
Rapeseed HOLL press cake	9.40 ± 0.38	32.20 ± 1.61	11.40 ± 1.14	6.80 ± 0.34	7.70 ± 0.39
Rapeseed classic press cake	7.83 ± 0.31	32.70 ± 1.60	10.90 ± 1.10	6.58 ± 0.33	6.48 ± 0.32
Sunflower high-oleic press cake	9.40 ± 0.38	28.20 ± 1.41	20.60 ± 0.21	6.42 ± 0.32	7.82 ± 0.39
Sunflower classic press cake	14.00 ± 0.56	24.60 ± 1.23	27.20 ± 0.27	5.60 ± 0.28	8.40 ± 0.42

**Table 2 foods-13-02969-t002:** Analysis of particle characteristics, including density, particle sizes, and distribution width, analysed in external accredited laboratories, and contact angles of different rapeseed and sunflower press cakes measured at Zurich University of Applied Sciences (n = 16). Measurements were performed for rapeseed press cake from HOLL rapeseed and conventional cultivation, rapeseed press cake from classic varieties and conventional cultivation, sunflower press cake from high-linoleic variety and conventional cultivation, and sunflower press cake from classic variety and organic cultivation. Results are given as mean values plus standard deviation.

	Density [g/cm^3^]	×10.0 [µm]	×50.0 [µm]	×90.0 [µm]	×10.0/×90.0 [-]	Contact Angle [°]
Rapeseed HOLL press cake	1.21 ± 0.22	2.96 ± 0.003	4.65 ± 0.012	9.96 ± 0.029	0.30 ± 0.001	78.91 ± 3.51
Rapeseed classic press cake	1.23 ± 0.22	3.15 ± 0.001	5.19 ± 0.002	10.52 ± 0.009	0.29 ± 0.0002	80.59 ± 6.98
Sunflower high-oleic press cake	1.23 ± 0.25	3.15 ± 0.011	5.34 ± 0.096	11.03 ± 0.113	0.29 ± 0.002	88.79 ± 2.65
Sunflower classic press cake	1.27 ± 0.23	3.09 ± 0.002	4.94 ± 0.021	10.71 ± 0.037	0.29 ± 0.001	90.58 ± 4.09

## Data Availability

The original contributions presented in the study are included in the article/Appendix A, further inquiries can be directed to the corresponding author.

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
