# Peer review of "Effect of Press Cake-Based Particles on Quality and Stability of Plant Oil Emulsions"

_foods, 2024, doi:10.3390/foods13182969_

Round 1

Reviewer 1 Report

Comments and Suggestions for Authors

Journal: Foods

Title: Effect of Press Cake-Based Particles on Quality and Stability of Plant-Oil Emulsions

The authors' ideas are good, but the experiments are incomplete and not suitable for publication in its current form.

-Graphical abstract

Authors should include the Graphical abstract that define the work was carried out and outcome of research. During research what are all the methodology adopted and developed things need to be incorporated in the graphical abstract. Please include it.

-Introduction

1. The reasons for selecting sunflower and rapeseed oils and their press cake pellets as the subject of study should have been highlighted in the introduction.

2.The introduction lacks logic. There are too many paragraphs and it is recommended to merge some of them.

- Material & methods section

1. Materials and chemicals used in the experiments should be listed in a separate section.

2. Authors should include the purity of each chemical utilized, as well as the details of each equipment used throughout the study.

3. Authors should discuss in detail the methodology adopted and procedure followed for conducting this study correlate with previous study. Please reshape it.

-Result and discussion section

1. In this section authors should provide detailed information regarding finding and outcome of research with numerical data and correlate it with previously published studies. The introductory sentences need to be avoided. Please reshape it.

2. Table 1: Data recorded as “mean ± standard deviation”.

3. Provide the original photographs taken during the contact angle measurement.

4. In Section 3.2, provide the appearance of the emulsion samples after different storage times.

5. For emulsion systems, it is very important to observe the interfacial structure. Therefore, it is recommended to supplement the CLSM pictures.

6. The authors only tested the viscosity change of the emulsions. It is recommended that the authors add rheological tests such as frequency scans and strain scans.

- References

1.Authors may ensure that they have adhered to the journal's referencing style throughout.

2. Eliminate old references from the manuscript and replace them with recent references. Use references from 2018-2024.

Comments on the Quality of English Language

The language of the manuscript needs polishing.

Author Response

1         Reviewer Report 1

The authors' ideas are good, but the experiments are incomplete and not suitable for publication in its current form.

Dear reviewer 1

Many thanks for the careful reading and critical feedback on our manuscript. We have added data where available but cannot fulfil all requests. The underlying reason is that the presented findings were part of a three-year research project with industrial partners and, hence, limited to more applied analyses that are highly relevant for the end application in palm oil replacement. In this type of project, fundamental analyses are out of scope. We are nevertheless convinced that the results are worth publishing as they highlight the possibilities and limitations of the use of natural particles from side streams in emulsion stabilisation.

Please find below a point-by-point answer to your comments and attached the manuscript in tracking mode. 

With kind regards,
The authors.

1.1         Graphical abstract

Comment 1: Authors should include the Graphical abstract that define the work was carried out and outcome of research. During research what are all the methodology adopted and developed things need to be incorporated in the graphical abstract. Please include it.

Answer 1: We have added more details and rearranged the graph to make it better understandable.

1.2         Introduction

Comment 2: The reasons for selecting sunflower and rapeseed oils and their press cake pellets as the subject of study should have been highlighted in the introduction.

Answer 2: We have added the following sentence to the introduction where we introduce the aim of the study: Rapeseed and sunflower seeds as the top two oilseeds grown in Switzerland are the raw materials of choice.

Comment 3: The introduction lacks logic. There are too many paragraphs and it is recommended to merge some of them.

Answer 3: We have rearranged the paragraphs and adjusted some introductory sentences.

1.3         Material & methods section

Comment 4: Materials and chemicals used in the experiments should be listed in a separate section.

Answer 4: All oils, press cakes and emulsifiers are now listed in a separate section with more detailed information on the producer.

Comment 5: Authors should include the purity of each chemical utilized, as well as the details of each equipment used throughout the study.

Answer 5: We have now listed all producers and variety details on the oils and press cakes. As for food-grade products such as the emulsifiers no purity is given, we could not add a degree of purity.  Details of all equipment are given in the original manuscript, and we have, hence, not added any further information on the type of equipment used.

Comment 6: Authors should discuss in detail the methodology adopted and procedure followed for conducting this study correlate with previous study. Please reshape it.

Answer 6: We have completely re-structured the material and methods section and focused on making the experimental design clear. We hope that the new structure meets your approval and is helpful for the reader. Concretely, we now separated the following sections: materials, production methods, analysis methods, experimental design and have made the details of the experimental design clear which is a sequence of unifactorial designs.  

We have furthermore added two former publications on particle stabilised foam which built the basis for decision for the particle analyses and on palm-fat replacement in baked goods which built the basis for the emulsification process.

1.4         Result and discussion section

Comment 7: In this section authors should provide detailed information regarding finding and outcome of research with numerical data and correlate it with previously published studies. The introductory sentences need to be avoided. Please reshape it.

Answer 7: Regarding numerical data, we have compared our own findings to previous studies in a qualitative and where possible quantitative manner. However, as the setting of the previous studies differs in key elements (different machinery used, different inner phase fractions, etc.), a numerical comparison is difficult. Regarding the presentation of the results, we had to decide for either a graphical display or numerical data (table format). We believe that the current form showing boxplots (incl. median, 25% and 75% quantiles) plus the compact letter display to highlight where group means differ statistically significantly (based on Wilcoxon post-hoc tests) is better legible for the reader. If desired, we are willing to hand out the original test-results in table format or as a supplementary table.

With respect to introductory sentences, we have carefully checked them again and removed most of them. We believe that the remaining text for section 3.3.4 is necessary for the reader to be able to follow the arguments w.r.t the results presented.

Comment 8: Table 1: Data recorded as “mean ± standard deviation”.

Answer 8: The analyses were all done with accredited laboratories which were in the past unwilling to give out standard deviations but have now finally agreed to share the data with us which we have added to the table.

Comment 9:  Provide the original photographs taken during the contact angle measurement.

Answer 9: Unfortunately, the photographs were not saved as the data volume was too high. We have however included below one exemplary image that shows a measurement of the contact angle.

Comment 10: In Section 3.2, provide the appearance of the emulsion samples after different storage times.

Answer 10: The volumes of the cream, emulsion and serum layers were recorded, but no images were taken. Therefore, we are unable to provide the requested images. As the project is closed and the raw materials discarded, we cannot redo the trials either.

Comment 11: For emulsion systems, it is very important to observe the interfacial structure. Therefore, it is recommended to supplement the CLSM pictures.

Answer 11: As no CLSM pictures were taken and the project is finished and the raw materials discarded, we can not provide this additional information.

Comment 12: The authors only tested the viscosity change of the emulsions. It is recommended that the authors add rheological tests such as frequency scans and strain scans.

Answer 12: Frequency sweeps and strain sweeps were not part of the project scope and we can, hence, not provide this data.

1.5         References

Comment 13: Authors may ensure that they have adhered to the journal's referencing style throughout.

Answer 13: The references were carefully double-checked and adapted to the referencing style. Concretely, we have adapted the numbering from an alphanumerical order to numbering the references according to the order the literature is mentioned in the text. Also, in the text we have now used square brackets and numbers instead of authors and publication year. Lastly, we have double checked all references in the reference list and adapted to the journal style.

Comment 14: Eliminate old references from the manuscript and replace them with recent references. Use references from 2018-2024.

Answer 14: we have carefully re-checked the importance of the older references and are convinced that they are of high importance for our discussion.

Reviewer 2 Report

Comments and Suggestions for Authors

1.       What is the main component of the press cakes for the emulsion stabilization? What is the primary quality (contact angle, size distribution, shape and so on) of the press cakes particles for O/W and W/O emulsions stabilizations?

2.       What are characterizations of the microstructure properties and zeta potentials of the press cakes particles? What are the potential applications and advantages of these particles compared with the other commercial-grade or reported Pickering particles?

3.       The morphology observation of the obtained W/O and O/W emulsions should be provided and compared. What are the stabilization mechanisms of the fabricated emulsions stabilized by these particles with or without additional surfactants? You should explain.

4.       The changes of droplet size and size distribution for emulsions during different storage time also should be presented and discussed.

5.       In Table 1 and Table 2, the significant difference analysis for data is needed.

6.       In Results and discussion section, please try to enrich discussion with more comparison to the past studies and theoretical information.

Comments on the Quality of English Language

Authors should check the manuscript carefully for the spelling and grammatical errors. The sentence tense for the manuscript needed to be corrected carefully.

Author Response

1         Reviewer Report 2

Dear Reviewer 2

Many thanks for your valuable feedback that we have addressed point by point below and attached the manucsript with all changes in tracking mode. We hope that our answers are to your satisfaction.

Kind regards,

The authors

Comment 1: What is the main component of the press cakes for the emulsion stabilization? What is the primary quality (contact angle, size distribution, shape and so on) of the press cakes particles for O/W and W/O emulsions stabilizations?

Answer 1: All these aspects (contact angle, size distribution, shape, density …) play together. The contact angle determines whether it is even possible to stabilise emulsions which is the case for the particles used. The size distribution is important with respect to the droplet size that can be stabilised. The shape is of lesser importance as a variety of shapes (round, elongated, fibrous, …) can lead to interfacial stabilisation. We have, nevertheless, added images of the different particles in a new, additional figure (Fig. 1) and included the effect of particle shape in our discussion. In addition to the classic particle qualities usually discussed, the press cake particles with their high content of dietary fibre furthermore led to an increase in the viscosity of the aqueous phase.

Comment 2: What are characterizations of the microstructure properties and zeta potentials of the press cakes particles? What are the potential applications and advantages of these particles compared with the other commercial-grade or reported Pickering particles?

Answer 2: Microstructure properties were measured and are summarised in table 2. Regarding the choice of particles: As the project scope was limited to the use of particles from oil press cake and the partners selected the oil and press cakes from sunflower and rapeseed oil, there was no option to use commercial-grade particles. Advantages of using press-cake particles of the same oilseed are: i) sustainability of the solution by using a side stream, ii) ease of declaration (no E-numbers), iii) sales argument (all natural, sustainable).

Comment 3: The morphology observation of the obtained W/O and O/W emulsions should be provided and compared. What are the stabilization mechanisms of the fabricated emulsions stabilized by these particles with or without additional surfactants? You should explain.

Answer 3: We have added a section on the combined stabilisation of interfaces using emulsifier plus particles to the introduction section. A detailed morphological observation of the W/O and O/W was however not performed as the turbidity of the emulsions caused by the soluble parts of the particles made a detailed visual assessment of the droplet interfaces impossible.

Comment 4: The changes of droplet size and size distribution for emulsions during different storage time also should be presented and discussed.

Answer 4: despite large effort to develop image analyses further to allow the analyses of the type of emulsions produced in this project, the success was limited due to the opaque areas formed by larger press cake particles and soluble components of the particles as well as by additional components such as waxes.

Comment 5: In Table 1 and Table 2, the significant difference analysis for data is needed.

Answer 5: Unfortunately, the laboratories originally only provided us with the median values of their measurements. We were now able to receive the standard deviations but have no access to the laboratories original raw data. Hence, we can only provide a significant difference analysis for the contact angle measurements done in-house (see results now mentioned in Sec. 3.1 below Tab. 2, and Tab. 1 below for your reference). This is why we have been careful in our discussion of the effect of differences in particle quality and have, e.g., not made any correlation of emulsion quality with particle density (where the standard deviations are large) and stated clearly that only one of the press cakes differed to a degree of practical significance in its oil content.

Table 1: Results of the Wilcoxon pairwise post-hoc test w.r.t. the contact angle measurements in the groups RH = Rapeseed HOLL press cake, RC = Rapeseed classic press cake, SHo = Sunflower high oleic press cake, SC = Sunflower classic press cake. U is the Mann-Whitney U statistic, eta-square is the effect and reject stands for rejecting the null-hypothesis at the 95% confidence level (alpha=0.05).

A

B

U

p

eta-square

reject

RH

RC

87.0

0.126875

0.021238

False

RH

SHo

0.0

0.000002

0.702736

True

RH

SC

0.0

0.000002

0.687391

True

RC

SHo

26.0

0.000130

0.361082

True

RC

SC

23.0

0.000082

0.416946

True

SHo

SC

94.5

0.213387

0.059673

False

RH = Rapeseed HOLL press cake, RC = Rapeseed classic press cake, SHo = Sunflower high oleic press cake, SC = Sunflower classic press cake

Comment 6: In Results and discussion section, please try to enrich discussion with more comparison to the past studies and theoretical information.

Answer 6: we have researched further and added additional discussion based on past studies. As the comparability to other studies is limited (different emulsification setup an parameters, different phase fractions, different type of particles and especially nearly no studies with natural raw materials in particle form as interface stabilisation), we have limited our discussion to studies that are comparable in at least some aspects but left out literature where we do not see sufficient comparability.

Reviewer 3 Report

Comments and Suggestions for Authors

The authors evaluate the effect of press cake on the stability of emulsions.

The authors use the term emulsion index %. It is not well defined in the manuscript on how you evaluated this parameter. Please explain in more details this important parameter. 

The authors use multiple symbols (a, b, c, d, and e) in the figures and also use a combination of these symbols. Please define these variables. 

In figure 5 there is a comparison between contact angle and viscosity. How did you do this experiment? 

Why did you pursue stability for 7 days? What is the objective of the emulsion preparation and how these emulsions will be used? 

Author Response

1         Reviewer Report 3

Dear Reviewer 3

Many thanks for your valuable feedback that has shown us where our writing was not clear enough. Please find below our point by point answers to your comments and attached the manuscript with all changes in tracking mode. We hope that our answers are to your satisfaction.

Kind regards,

The authors

Comment 1: The authors evaluate the effect of press cake on the stability of emulsions. The authors use the term emulsion index %. It is not well defined in the manuscript on how you evaluated this parameter. Please explain in more details this important parameter.

Answer 1: we have adjusted the text as follows and hope it is clearer now:

The emulsion stability was determined by filling 14.5 ml of the prepared emulsions into 15 ml Falcon tubes and storing them at 20 °C. The volumes [ml] of the cream layer (oil phase), the emulsion layer (emulsion), and the serum layer (water phase) were determined relative to the initial volume [ml] via visually assessing the ml scale of the tubesat 1 h, 24 h and 168 h after the emulsion formation, according to the protocol described by McClements (2007). The emulsion index  [%] is defined by

(1)

where  is the total volume directly after emulsion formation and  is the volume without any phase separation after storage of t = 1 h, 24 h, 168 h, respectively.

Lab-scale trials were conducted in duplicates, with each trial analysed three times (n=6). In comparison, pilot-scale trials were conducted in triplicates and analysed three times each (n=9).

Comment 2: The authors use multiple symbols (a, b, c, d, and e) in the figures and also use a combination of these symbols. Please define these variables.

Answer 2: to avoid repetition in the manuscript, we have only explained the meaning of the letters in the materials and methods section and not repeated it for each graph. We have extended the explanation in the materials and method section to: Significant differences were indicated with different letters (compact letter display) where only results with no overlap in letters differ significantly from each other (e.g. a and b are significantly different, a and ab are not significantly different).  

Comment 3: In figure 5 there is a comparison between contact angle and viscosity. How did you do this experiment?

Answer 3: Since the different press cake particles have different properties (like contact angle), the influence of these properties on the emulsion properties (like viscosity) can be investigated through their correlation coefficients.

Comment 4: Why did you pursue stability for 7 days? What is the objective of the emulsion preparation and how these emulsions will be used?

Answer 4: A maximal stability would be preferred but longer storage times were critical with respect to mould growth. A solution to this would be a heat treatment of the press cakes before use which is envisaged for industrial application.

Round 2

Reviewer 1 Report

Comments and Suggestions for Authors

The paper has well revised, and suggest to accept.

Author Response

Reviewer 1 comment 1: Graphical abstract is not a requirement in this journal, is it. If so, the authors did not address this comment.

Answer 1: As discussed with Dr. Romruen, this issue is solved.

Reviewer 2 comment 5: It is not satisfactory to say that you were only provided the median (I presume the authors meant "mean") scores and standard deviations of the data. Especially, Table 2. Mean separation analysis should have been proactively requested for if the analysis was not done by the author. And again, raw data should have been provided to the authors. This is especially important in the context of data integrity. This is enough to reject this paper. However, the subject addressed, and the body of work are important for us to go that route. We strongly suggest to the authors to ensure they secure all their raw data in the future and not leave to the lab that collected them.

Answer 2: We fully agree and were astonished how difficult it was to receive the desired information on the standard deviations from the laboratories. In the future, we will ask the laboratories to provide the raw data to us from the beginning on and hope for more success with receiving said data then.

We have changed to the word mean where applicable (i.e. most values from external laboratories).  However, for the median particle size we have left the word median as X50,0 is the median value and not the arithmetic mean.  

Reviewer 3 comments 2: As a rule of thumb, you MUST define acronyms, symbols, and superscripts/subscripts used in every table and figure.... This is especially important where similar letters or symbols are used for different purpose. Context MUST be provided each time. in Figure 5, letters were used to indicate significant difference between mean of values in the plots without defining what they are and in what context they were used. There is a need to correct this by clearly defining what these letters stand for in this figure; and in do the same for every figure or table.... using phrase like: "letter(s) on every box-pot that are different indicate significant difference with the rest at P < 0.0X"... something like that... The ground of avoiding repetition does not hold here. This type of repetition is good clarity... Please, correct throughout the manuscript.

Answer 3:  Thank you for pointing this out. We have added the following sentence to each caption: Letters indicate groups of significance, i.e., group means not sharing any letter are significantly different by the Wilcoxon test with a p-value of 0.05.

Reviewer 2 Report

Comments and Suggestions for Authors

The manuscript has been greatly improved and several problems have been well corrected. Overall, it can be accepted for the publication.

Author Response

Thank you for the positive feedback on our changes.